# Tracking Cognitive Development of Large Language Models

## Abstract

Large Language Models (LLMs) have recently shown tremendous performance on a large variety of Natural Language Processing tasks, ranging from text comprehension to mathematical problems. However, the mechanism regarding why and how such performance has been achieved remains unknown, and it is unclear whether LLMs can achieve human-like cognitive abilities or whether these models are still fundamentally limited. To bridge this gap, we introduce Piaget's Theory of Cognitive Development (PTC) as a tool to reveal the development of cognitive abilities of LLMs. We construct a benchmark (CogLM) based on the scenario experiments in PTC to evaluate the cognitive level of LLMs, covering 10 abilities and 1220 questions created by more than 20 human experts. Through extensive experiments across multiple LLMs on CogLM, we find that: (1) Human-like cognitive abilities have emerged in State-of-the-art LLMs (GPT4), comparable to those of 20-year-old humans. (2) The parameter size and optimization objective are two key factors affecting the cognitive abilities of LLMs. (3) The ability of downstream tasks highly depends on the level of cognitive abilities. These findings provide guidance for the future development of advanced abilities of LLMs from the perspective of ability evolution, and shed light on the mystery behind the emergence of advanced abilities of LLMs.

## 1 Introduction

Large Language Models (LLMs) have recently achieved impressive performance on a large variety of Natural Language Processing (NLP) tasks, including text comprehension (Kenton & Toutanova, 2019), reasoning (Talmor et al., 2020; Webb et al., 2023), code generation (Chen et al., 2021), and mathematical problems (Fu et al., 2023). However, there is little theoretical evidence regarding why and how this performance has been achieved. While recent research focuses on evaluating or enhancing the upper bound capabilities of LLMs on certain type of tasks (Valmeekam et al., 2022; Gendron et al., 2023), few studies explore the reason behind evolutionary relationship between different abilities. The lack of theoretical explanations for the emergence of capabilities in LLMs will become an obstacle on the path towards artificial general intelligence (AGI).

We aim to measure the cognitive abilities of LLMs, so as to better understand the current LLMs' performance on downstream tasks and shed light on the future improvement of LLMs' advanced abilities. Inspired by the recent success of cognitive psychology research on LLMs (Binz & Schulz, 2023; Kosinski, 2023), we introduce Piaget's Theory of Cognitive Development (PTC) (Piaget et al., 1952; Flavell, 1977; Badakar et al., 2017) as a tool to evaluate the cognitive abilities of LLMs. As the most authoritative theory in the development of psychology, PTC suggests that human children move through four different stages of learning, including the sensorimotor stage (0-2 years old), the preoperational stage (2-7 years old), the concrete operational stage (7-12 years old), and the formal operational stage (above 12 years old). Children in different cognitive stages have significantly different patterns of thinking and abilities of cognition, and will gradually transition to the next stage with the acquisition of necessary abilities. The natural questions are: At what stage has the cognitive ability of LLMs developed compared to humans at present? What are the key factors that affect the cognitive abilities of LLMs? Is the emergence of advanced abilities and performance bottlenecks in current LLMs related to their cognitive levels?

To better explore the above questions, we construct a benchmark based on the scenario experiments used in PTC for evaluating the cognitive abilities of LLMs, denoted as CogLM [1]. To ensure the alignment between CogLM and PTC, we conduct human testing with 207 individuals aged 6-20 years old. We then perform extensive experiments on CogLM over several series of language models, including OPT (Zhang et al., 2022), LLaMA 2 (Touvron et al., 2023), ChatGPT and GPT4(OpenAI, 2023). Our results indicate that: (1) Human-like cognitive abilities have emerged in State-of-the-art LLMs (GPT4), comparable to those of 20-year-old humans. (2) The parameter size and optimization objective are two key factors affecting the cognitive abilities of LLMs.

We believe that our findings can provide a novel insight into the emergence of abilities in LLMs, and shed light on the future development of advanced abilities of LLMs. Our contributions can be summarised as follows:

- We innovatively introduce PTC to reveal the development of cognitive abilities of LLMs.
- We construct a high-quality benchmark (CogLM) for evaluating the cognitive ability level of LLMs.
- We evaluate multiple LLMs on CogLM, and show that the ability of downstream tasks highly depends on the level of cognitive abilities.

## 2 RELATED WORK

**LLM Evaluation**   Due to the importance of LLMs, their abilities have been thoroughly evaluated on a wide range of problems. Large-scale efforts have been invested in constructing large benchmarks itegrated with numerous LM evaluations across a number of fields (Srivastava et al., 2022; Liang et al., 2022; Hendrycks et al., 2020). Due to the extremely superior performance of LLMs in a number of traditional NLP tasks, recently challenging tasks have been proposed to test the upper bound performance of LLMs (Hendrycks et al.; Valmeekam et al., 2022; Gendron et al., 2023). While previous benchmarks focus on measuring either a type or a category of advanced ability in LLMs, few studies explore the development relationship between different abilities, which is crucial for understanding the emergence of LLMs' abilities.

**Cognitive psychology survey on LLMs**   Several works introduce tools from cognitive psychology to study LLMs. Such as understanding the behavior in LLMs (Ritter et al., 2017; Kosoy et al., 2022; Hagendorff et al., 2022), exploring the human-like abilities in LLMs (Han et al., 2022; Kosinski, 2023; Aher et al., 2023; Pan & Zeng, 2023), and improving LLMs' performance on certain task (Betz et al., 2021). Our work is most similar to present work on using cognitive psychology to explore whether LMs "learn and think like people" by Binz & Schulz (2023), which suggests that LLMs struggle to reason causally due to the differences in how humans and LLMs learn about the world. The key difference in our approaches is that Binz & Schulz (2023) aims to study GPT-3 by assessing its advanced abilities (e.g. decision-making, information search, deliberation, and causal reasoning), while we evaluate the cognitive abilities of multiple LLMs at different stages to track the evolution process of their cognitive abilities.

**Piaget's Theory of Cognitive Development**   Theory of Cognitive Development (PTC) is the most authoritative theory in the development of psychology, developed by Jean Piaget (Piaget et al., 1952). PTC suggests that intelligence grows and develops through a series of stages. As childs interact with the world around them, they continually add new knowledge, build upon existing knowledge, and adapt previously held ideas to accommodate new information. PTC is widely used in education, psychology, linguistics, and neuroscience, providing a theoretical framework and methodology for research in these areas.

## 3 COGLM BENCHMARK DEVELOPMENT

To comprehensively and accurately assess the cognitive abilities of LLMs, we undertake the following efforts: (1) We revisit 12 cognitive abilities of PTC, 10 of which are selected and redefined to

---

[1]Considering the value of the data, we will release our data and code as soon as the paper is accepted.

construct CogLM according to the characteristics of LLMs (section 3.1). (2) We create standardized data construction guidelines to ensure the quality of CogLM (section 3.2). (3) We conduct extensive human testing to ensure the alignment between CogLM and PTC (section 3.3). (4) We build a Calibrated Mapping Function to establish a reliable mapping between testing results on CogLM and cognitive age (section 3.4).

## 3.1 DEFINITION OF COGNITIVE ABILITIES

According to PTC, the development of human cognition is divided into four stages, which include 12 cognitive abilities. Considering that the interaction interface of most LMs is limited to text-based format, we exclude *reflexes* and *sensorimotor* aspects of multimodal interaction and build CogLM based on the remaining 10 cognitive abilities.

We strictly define the concept of each cognitive ability based on PTC and provide representative examples for explanation as shown in Table 1.

## 3.2 STANDARDIZED ANNOTATION GUIDELINES

To ensure that CogLM can accurately reflects the cognitive abilities of LLMs, we have established standardized annotation guidelines and strictly adhere to them during the annotation phase:

**Data Format** Although modern LLMs generally possess strong generation capabilities, early-aged LLMs (e.g., GPT-2) have limited generation abilities (similar to Human infants). Therefore, we have opted for multiple-choice questions as the assessment format. This approach avoids the influence of variations in generation capabilities on the accurate evaluation of cognitive abilities.

**Number of Samples** Abilities in the early stages are relatively simple and have a more concentrated form of expression, while abilities in the later stages are more comprehensive and have a more diverse form of expression. Based on this, we have set the number of samples to increase with each stage, as shown in the Table 2.

**Qualified Annotator** We select adults with backgrounds in psychology or artificial intelligence as data annotators. Annotators are provided with comprehensive materials on PTC and required to study them carefully. We then assess annotators' understanding of PTC through exams (see Appendix Table 10 for the examination paper). Finally, we provide annotators with at least three example samples for each cognitive ability. Each annotator is required to annotate no fewer than 30 questions and options for two specific cognitive abilities.

**Annotation Quality Control** After annotation, we conduct cross-checks among annotators to identify samples with quality issues. Quality issues include questions that cannot effectively assess the corresponding cognitive abilities, questions with ambiguities, and elements of bias or violence.

## 3.3 CONSISTENCY WITH THEORY

After the dataset construction is completed, we consider conducting human tests to further ascertain whether CogLM is consistent with PTC and whether it can effectively reflect cognitive abilities. We randomly select 10 samples from each subset of CogLM to create questionnaires, which are then distributed to testers aged between 6 and 20. Out of the 207 completed questionnaires, 141 are deemed valid (based on the reasonableness of test duration). We calculate the Spearman and Pearson correlation coefficients between the age of the participants and their questionnaire scores. It turns out that spearman correlation is 0.7169 and pearson correlation is 0.7362 (all the p-values $< 1e-10$), indicating a strong correlation between them. This statistical result validates the effectiveness of the Standardized Annotation Guidelines we have developed in ensuring the efficacy of CogLM for assessing cognitive abilities.

## 3.4 CALIBRATED MAPPING FUNCTION

After confirming the positive correlation between answer accuracy and cognitive age, we aim to further construct the mapping function between them. We first make adjustments to the method of calculating accuracy. The number of candidate options for questions in CogLM falls within the range $[2, 5]$. Such a variability can impact the likelihood of providing a correct answer through

| |
|---|
| **First Stage:** *Constancy* (***const***) |
| **Definition:** Objects exist both within and outside the field of vision and maintain a certain level of stability. |
| **Example:** Q: Assuming there is a small ball on the table. Is the ball still on the table when covered with a cloth? Ans: Yes |
| **First Stage:** *Early Representation* (***early***) |
| **Definition:** Objects are endowed with corresponding meanings, thus gradually forming a universe of permanent objects. |
| **Example:** Q: How would you describe the color of snow? Ans: White |
| **Second Stage:** *Semiotic Function* (***semio***) |
| **Definition:** Use symbols to represent things and concepts. |
| **Example:** Q: Which item best represents love and romance? Ans: Rose |
| **Second Stage:** *Empathy* (***empat***) |
| **Definition:** Start considering others' perspectives and feelings when addressing issues. |
| **Example:** Q: You are fond of video games, but your cousin enjoys outdoor sports. What birthday gift would you give to him? Ans: Camping tent |
| **Third Stage:** *Reversibility* (***rever***) |
| **Definition:** Understand the reversibility of physical operations and is capable of reverse thinking. |
| **Example:** Q: If a plane will land at 10 AM and fly for 6 hours, what time will it take off? Ans: 4 AM |
| **Third Stage:** *Conservation* (***conse***) |
| **Definition:** The external alteration of forms doesn't affect certain fundamental attributes. |
| **Example:** Q: If a stone is cut into two, what will be their total mass? Ans: Not change |
| **Third Stage:** *Induction* (***induc***) |
| **Definition:** Infer universal rules based on observed results. |
| **Example:** Q: Given an arithmetic sequence: 2, 5, 8, 11, 14, which of the following is the general term formula for this sequence? Ans: 3n + 2 |
| **Forth Stage:** *Hypothetico-Deductive* (***deduc***) |
| **Definition:** Deduce practical problems based on specific assumptions or rules. |
| **Example:** Q: Alex is excited, Paul is sad, Mike is crying, Anna is angry. The sad one is dog, the angry one is swan, the crying one is cat, the excited one is tiger. Swan likes cat, cat likes tiger, tiger likes dog, dog likes swan. What does Anna like? Ans: Cat |
| **Forth Stage:** *Propositional Operation* (***propo***) |
| **Definition:** Understand propositions and determine the logical relationships between propositions. |
| **Example:** Q: Sentence1: In fact, the Lions of Delos were made from Naxos marble. Sentence2: There are five Lions of Delos, and also two Tigers of Delos. What is the propositional relationship between sentence1 and sentence2 ? Ans: Neutral |
| **Forth Stage:** *Plan* (***plan***) |
| **Definition:** Develop sensible solutions based on specific problem. |
| **Example:** Q: Please plan an action execution sequence according to the rules. The following rules must be followed: going fishing before going hiking, doing yoga before going hiking, taking photos before doing yoga. Based on the above rules, please choose an action execution sequence that meets the rules from the following options to finalize: going hiking. Ans: taking photos, doing yoga, going fishing, going hiking |

Table 1: Definitions and examples of cognitive abilities included in CogLM.

guessing when participants are uncertain. Therefore, we calculate the calibrated accuracy on certain subset $\mathbb{S}$ as follows:

$$Acc = \frac{1}{|\mathbb{S}|} \times \sum_{i=1}^{|\mathbb{S}|} \frac{\mathbf{1}_{\text{predict}_i = \text{answer}_i} - 1/|\text{candidates}_i|}{1 - 1/|\text{candidates}_i|} \tag{1}$$

A negative calibrated accuracy (worse than random selecting) indicates a clear deficiency in the corresponding cognitive ability. We further use 80% of the questionnaire results in Section 3.3 as

Table 2: Data statistics on all ability subsets of CogLM.

| CogLM | stage 1 | | stage 2 | | stage3 | | | stage4 | | | Overall |
|---|---|---|---|---|---|---|---|---|---|---|---|
| | const | early | semio | empat | rever | conse | induc | deduc | propo | plan | |
| Sample Number | 50 | 100 | 100 | 100 | 100 | 110 | 100 | 250 | 100 | 210 | 1220 |
| Question Tokens | 18.5 | 11.36 | 11.55 | 25.27 | 30.0 | 26.3 | 42.0 | 51.8 | 30.0 | 77.9 | 39.5 |
| Candidates Number | 2.00 | 4.00 | 3.96 | 2.96 | 4.00 | 2.98 | 4.00 | 4.00 | 3.00 | 4.00 | 3.66 |
| Candidates Tokens | 1.00 | 1.19 | 1.48 | 4.23 | 3.87 | 4.28 | 7.58 | 1.00 | 1.00 | 20.30 | 5.71 |

the training set $\mathbb{S}_Q$ to optimize the regression function $f(\cdot)$ as follows:

$$\mathcal{L}_{regression} = \frac{1}{|\mathbb{S}_Q|} \times \sum_{i=1}^{|\mathbb{S}_Q|} (f(Acc_i) - age_i)^2$$

$$f(Acc) = \sum_{i=1}^{4} w_i \times Acc_{\text{stage}i} + b$$

(2)

The Spearman correlation between the age predicted by $f(\cdot)$ and the real age on the remaining 20% samples is 0.9354, which signifies that $f(\cdot)$ can precisely approximate the mapping from results on CogLM to cognitive age. We observe that $w_1 : w_2 : w_3 : w_4 = 1 : 2.6 : 1.4 : 2.5$, indicating that cognitive abilities in the second and fourth stages are better at reflecting cognitive age under the evaluation of CogLM.

## 4 EXPERIMENTS

### 4.1 EXPERIMENT SETUP

**Models**   As shown in Table 3, We perform evaluations on the most recent and popular architectures for NLP tasks. We restrict our experiments to Large Language models (LLMs). We conduct experiments on the popular family of GPT architecture: OPT series (Zhang et al., 2022), including models with sizes of 1.25M, 1.3B, 2.7B, and 6.7B, optimised for text completion; GPT-3.5-Turbo (commonly referred to ChatGPT), optimised for chat; and GPT4, for which the training and architecture details are unknown (OpenAI, 2023). We also perform experiments on LLaMA2 family of models(Touvron et al., 2023), including models with scale of 7B, 13B and 70B. In particular, LLaMA2-text models are pretrained generative text models for text completion, while LLaMA2-chat is fine-tuned variation optimised for dialogue use cases.

Table 3: The statistics of considered language models.

| Type | Series | Size |
|---|---|---|
| Text completion | OPT | 125M, 1.3B, 2.7B, 6.7B |
| | LLaMA2-text | 7B,13B,70B |
| Chat completion | LLaMA2-chat | 7B,13B,70B |
| | GPT-3.5-Turbo | N/A |
| | GPT4 | N/A |

**Evaluation**   For ChatGPT and GPT-4, we use the Open AI API to run all the evaluations. As ChatGPT and GPT-4 are chat-completion models, we provide the instructions in chat format. For OPT, LLaMA-2-text and LLaMA-2-chat series models, we use the weights provided on the Huggingface hub. LLaMA-2-chat models are used as chat-completion models, while the others are used as text-completion models. For text-completion models, as they lack the ability to follow instructions and their output format is difficult to control, we concatenate each option with the corresponding question as input, and take the option with the highest generation probability as the model's prediction.

For chat-completion models, we constrain the format of the model's generated answers through instructions. [2] We consider a model to provide a valid answer even if the format is incorrect. Unless specified otherwise, we always ask the model to provide a single answer with explanations.

**Sampling settings**    When the chat-completion models are queried for completions, we set the temperature parameter $T$ as 0 and $p$ in nucleus sampling as 1, leading to deterministic answers. Unless otherwise specified, we use the default settings.

## 4.2    MAIN RESULTS

As shown in Table 4, We run the model with the largest number of parameters in each series on cogLM, and report the adult human performance for comparison. Overall, the cognitive abilities of the OPT, LLaMA2-chat 70B, GPT-3.5-Turbo, and GPT4 models successively increase, and the performance of each model gradually declines with the increase of stage, consistent with humans. Specifically, The current state-of-the-art model (GPT-4) has emerged with strong cognitive abilities, reaching the level of a 16-year-old human. It is also worth noting that both GPT-3.5-Turbo and GPT-4 surpass humans in empathy ability at stage 2, which is natural, as humans tend to have some degree of selfishness. Despite its superior performance, GPT-4's performance on plan ability (59.4) is still barely satisfactory, far behind that of humans (95.6), which is consistent with the conclusion of Valmeekam et al. (2022). Our results indicate that enhancing the ability of planning is the major direction for improving the overall cognitive abilities of LLMs in the future. For more detailed evaluation results, please refer to Appendix-Table 8.

Table 4: Calibrated accuracy (%) of largest model in evaluating series. Acc and Age refer to calibrated accuracy and the age of equivalent human performance. The value of Age is calculated according to Equation 2 and rounded to the nearest integer. Bold indicates the best performance.

| Model | stage1 | | stage2 | | stage3 | | | stage4 | | | Acc | Age |
|---|---|---|---|---|---|---|---|---|---|---|---|---|
| | const | early | semio | empat | conse | induc | rever | deduc | propo | plan | | |
| OPT 6.7B | -4.0 | 64.2 | 41.1 | -3.0 | 13.5 | 10.6 | 20.1 | -0.8 | -0.5 | 14.2 | 15.5 | 6.5 |
| LLaMA2-chat 70B | 52.1 | 96.2 | 78.5 | 66.2 | 68.4 | 65.3 | 44.0 | 15.2 | 40.0 | 20.6 | 54.6 | 14.1 |
| GPT-3.5-Turbo | 92.0 | 97.3 | 89.2 | 85.5 | 65.7 | 64.0 | 62.3 | 26.9 | 50.5 | 9.8 | 64.3 | 16.1 |
| GPT4 | 96.0 | 97.3 | 96.0 | **90.3** | 90.4 | 78.7 | 78.7 | 99.4 | 61.0 | 59.4 | 84.7 | 20.0 |
| Human | **100.0** | **98.0** | **96.1** | 84.2 | **98.2** | **91.6** | **92.0** | **100.0** | **82.0** | **95.6** | **93.7** | **21.5** |

## 4.3    ANALYSIS AND DISCUSSION

### 4.3.1    WHAT ARE THE KEY FACTORS AFFECTING THE COGNITIVE ABILITIES OF LLMS?

We explore this question from two perspectives: the parameter size and the optimization objective of LLMs, as they are the most natural assumptions. We leave the exploration of factors that require changes to the parameters of LLMs (e.g. fine-tuning on different types of datasets) for future work.

**The parameter size of LLMs**    As shown in fig 1, we compare the overall performance of models with different parameter size across OPT and LLaMA2-chat series, and report the performance of humans at different stages as a reference. Specifically, the cognitive abilities of LLMs continuously improve as the size of model parameters increases, which follows the same pattern as human cognitive abilities continuously improving with age. We hypothesize that the reason for this phenomenon is that the parameter size of LLMs can be analogous to a child's brain capacity, and cognitive abilities continuously improve with the development of the brain, as stated in Piaget et al. (1952).

**The optimization objective of LLMs**    As shown in fig 2, we compare the performance of LLaMA2-text 70B and LLaMA2-chat 70B at each stage on CogLM. The results show that the performance of both models generally declines with the increase of stage, while the performance of LLaMA2-chat 70B far exceeds that of LLaMA2-text 70B at every stage. Given that LLaMA2-chat

---
[2]We set the valid output format as: "The answer is $\boxed{option}$" in the prompt.

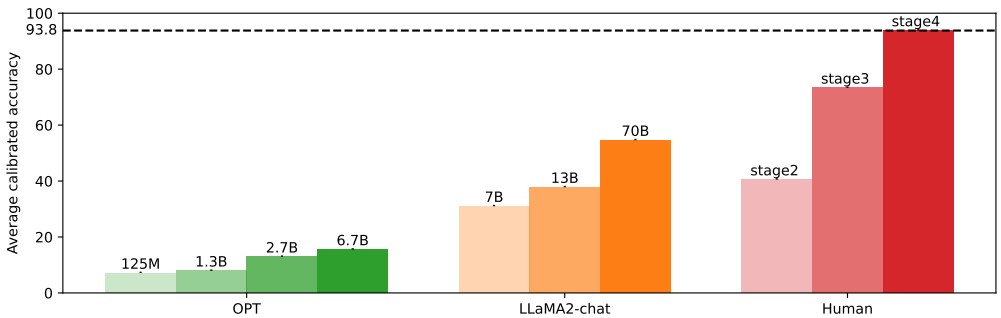

Figure 1: Average calibrated accuracy (%) of models with different parameter size and humans in different cognitive stage.

70B is further fine-tuned on dialogue data and RLHF trained compared to LLaMA2-text 70B, it suggests that LLMs could potentially enhance their cognitive abilities through learning to chat with humans, as RLHF is another approach for LLMs to learn the world, apart from text pretraining.

Based on the two sets of experiments above, we can draw the conclusion that the parameter size and optimization objective are key factors affecting the cognitive abilities of LLMs.

### 4.3.2 CAN ADVANCED TECHNOLOGIES HELP ENHANCE LLM'S COGNITIVE ABILITIES?

To answer this question, we applied two representative techniques separately to measure whether cognitive abilities of LLMs could be improved.

**Effect of Chain-of-Thought** The approach of guiding LLMs to subsequently solve problems has been shown to significantly enhance the performance in most scenarios (Wei et al., 2022). Thus, we are curious whether Chain-of-Thought (COT) can also be effective in improving the cognitive abilities of LLMs. We tested the performance of GPT-3.5-Turbo with and without COT separately on the CogLM and the results are shown in Table 5. From the perspective of the average calibrated accuracy of all the cognitive abilities, COT does not bring a significant performance improvement. We hypothesize that this is because cognitive abilities are inherent to the LLMs and cannot be enhanced through multi-step reasoning.

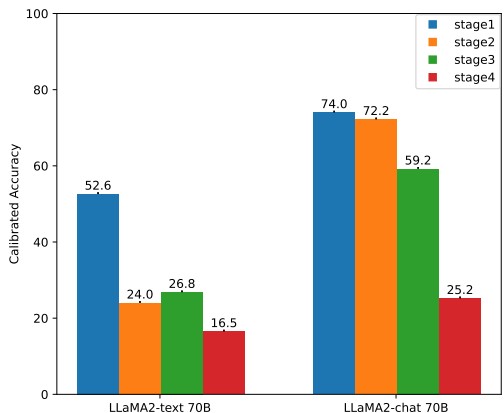

Figure 2: Comparison of the performance of LLaMA2-text 70B and LLaMA2-chat 70B at each stage on CogLM.

Table 5: Calibrated accuracy of GPT-3.5-Turbo on CogLM with multiple settings. "Base" indicates settings where both COT and SC are not used.

| Ability | const | early | semio | empat | conse | induc | rever | deduc | propo | plan | Avg |
|---|---|---|---|---|---|---|---|---|---|---|---|
| Base | **92.0** | 97.3 | **90.6** | 85.5 | **65.9** | 64.0 | 61.3 | 27.5 | 49.0 | 6.7 | 64.0 |
| *w/* COT | **92.0** | 97.3 | 89.3 | 85.5 | 65.7 | 64.0 | **62.3** | 26.9 | 50.5 | **9.8** | 64.3 |
| *w/* SC T=0.3 | 91.0 | **97.6** | 90.3 | **86.0** | 65.7 | 66.0 | 61.0 | **27.9** | **53.5** | 4.8 | **64.4** |
| *w/* SC T=0.7 | 91.0 | **97.6** | 90 | 85.5 | 65.7 | **66.7** | 61.3 | 27.7 | 52.0 | 3.5 | 63.1 |

**Effect of Self-Consistency** Self-Consistency (SC) (Wang et al., 2023) is another commonly used method that can effectively enhance the performance of LLMs. Multiple candidate predictions to a specific problem are suggested to generate through sampling following with a voting mechanism to eliminate noise introduced by single sampling. We conducted experiments with sampling times as 40 at temperature $T$ of 0.3 and 0.7, respectively. As shown in Table 5, similar to COT, SC can only bring about a very marginal improvement. This phenomenon is consistent with human. For example, for a boy who lacks the ability of empathy, no matter how many times he is asked to choose, he may find it difficult to realize that a scarf might be a more suitable gift for his grandmother than a lollipop.

Based on the two sets of experiments above, we can draw the conclusion that similar to human beings, it is challenging to achieve significant improvements in LLM's cognitive abilities without external stimuli.

### 4.3.3 HOW COGNITIVE ABILITY AFFECTS THE PERFORMANCE OF LLM

According to PTC, the development of human cognitive abilities is a gradual process, where the cognitive abilities of early stages can influence the advanced cognitive abilities. Additionally, cognitive abilities significantly determines the capacity to solve real-world problems. Therefore, we are very interested in whether these two aspects are similarly manifested in LLMs.

**The Interdependence Between Cognitive Abilities** Through preliminary experiments, we found that advanced LLMs' ability to follow instructions can help us erase specific cognitive abilities using a cognitive-ability-setting-prompt (e.g., "You have not yet formed a sense of empathy". See Appendix-Table 9 for all the prompts). On this basis, We investigated the interdependence of cognitive abilities in LLMs by selectively removing specific cognitive capabilities and testing them on CogLM. According to the experimental results shown in Figure 3, we can draw the following conclusions: (1) Advanced cognitive abilities significantly rely on early cognitive abilities, which indicates that the dependency relationships of LLM's cognitive abilities are similar to

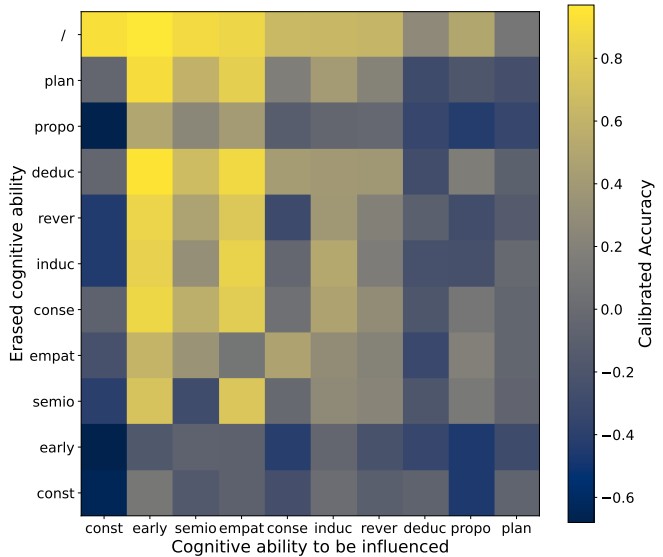

Figure 3: Cognitive ability interdependence matrix. The vertical axis represents cognitive abilities erased through prompts, and the color depth (calibrated accuracy) indicates the impact on the corresponding horizontal axis abilities after erasure.

those of humans. (2) The darker colors along the diagonal indicate that the way we erase the corresponding cognitive abilities is effective. (3) Constancy is a fundamental capability (in line with PTC), as it significantly influences and is influenced by other cognitive abilities.

**The Dependence of Downstream Ability on Cognitive Ability** In Table 4, we observed a gradual increase in cognitive abilities for OPT, LLaMA2, GPT-3.5-Turbo, and GPT4. On the other hand, based on extensive evaluation studies (Srivastava et al., 2022; Touvron et al., 2023; Liang et al., 2022), we also noted that this ranking result corresponds with the overall performance of LLMs when it comes to solving downstream tasks. This suggests that cognitive abilities are significantly correlated with practical skills for LLMs. To further understand this correlation, we conducted experiments to assess LLM's performance on downstream tasks when specific cognitive abilities were erased by cognitive-ability-setting-prompt. We chose representative math reasoning dataset GSM8K (Cobbe et al., 2021) and commonsense reasoning dataset StrategyQA (Talmor et al., 2019)

to conduct our experiments. As shown in Table 6, it is reasonable that the erasure of hypothetico-deductive, propositional operation and plan abilities significantly impact the performance of GPT-3.5-Turbo on GSM8K dataset as they are core abilities to solve math problems. We also found that the erasure of other cognitive abilities (especially in early stages) can also bring a strong impact, even if they may not seem helpful in solving math problems. Similar conclusions can be drawn on the StrategyQA dataset. These findings indicate that LLMs' abilities to solve downstream tasks strongly depend on the level of cognitive abilities. The advanced cognitive capabilities of GPT-3.5-Turbo and GPT-4 on CogLM partially account for their outstanding performance in various downstream tasks. From this perspective, we can further understand that Zero-shot COT (Kojima et al., 2022) is essentially enhancing LLMs' cognitive ability of deduction for improved performance on downstream tasks by incorporating "Let's think step by step" into the prompt.

Table 6: Accuracy (%) of GPT-3.5-Turbo on GSM8K and StrategyQA datasets when different cognitive abilities are erased.

| Erased Ability | const | early | semio | empat | conse | induc | rever | deduc | propo | plan | / |
|---|---|---|---|---|---|---|---|---|---|---|---|
| GSM8K | 0.1 | 0.6 | 25.5 | 16.6 | 38.6 | 30.7 | 21.2 | 12.1 | 1.0 | 2.2 | 59.9 |
| StrategyQA | 3.8 | 9.1 | 5.6 | 33.5 | 15.7 | 14.7 | 18.8 | 28.4 | 12.9 | 31.6 | 65.2 |

### 4.3.4 POTENTIAL APPLICATIONS OF ADVANCED LLM COGNITIVE ABILITY

Although there is still room for improvement, the cognitive abilities of advanced LLMs have approached levels close to that of adult humans as discussed in Section 4.2. A natural question is, what are the potential applications for advanced LLMs' cognitive abilities? When humans address cognitive questions, they deduce and provide answers based on their cognitive abilities. While we have demonstrated in Section 4.3.2 that the cognitive chain-of-thought (Chain-of-Cognition, COC) generated by LLMs barely help self-address cognitive questions, we are curious whether COC can assist early-aged LLMs in improving cognitive performance. On this basis, we use the question together with the COCs generated by advanced LLM (GPT-3.5-Turbo) as input to test the performance of early-aged LLM (LLaMA2-chat 7B) on CogLM. As shown in Table 7, in most cognitive abilities, COC can significantly improve the performance of LLaMA2-chat 7B. We leave the research on using COC from advanced LLMs to guide the improvement of cognitive abilities in early-aged LLMs and even children for future exploration.

Table 7: Calibrated accuracy of LLaMA2-chat 7B on CogLM with and without Chain-of-Cognition from GPT-3.5-Turbo as input.

| Ability | const | early | semio | empat | conse | induc | rever | deduc | propo | plan | Avg |
|---|---|---|---|---|---|---|---|---|---|---|---|
| $w/o$ COC | 32.0 | **78.6** | **74.5** | 54.9 | -0.2 | 41.3 | 25.3 | 6.1 | -2.0 | -0.1 | 31.0 |
| $w/$ COC | **54.2** | 57.3 | 53.3 | **69.4** | **34.1** | **41.9** | **34.7** | **8.1** | **26.9** | **1.5** | **38.1** |

## 5 CONCLUSION

Large Language Models perform very well on a large range of NLP tasks. Despite the great success, it is important to understand whether LLMs can achieve human-like cognitive abilities or not. In this paper, we introduce Piaget's Theory of Cognitive Development (PTC) as a tool to track the development of cognitive abilities of LLMs. We construct CogLM based on the scenerio experiments used in PTC, and conduct thorough human testing to ensure the alignment between CogLM and PTC. Through extensive experiments on multiple series of LLMs, we show that: (1) Human-like cognitive abilities have emerged in State-of-the-art LLMs (GPT4), comparable to those of 20-year-old humans. (2) The parameter size and optimization objective are two key factors affecting the cognitive abilities of LLMs. (3) The ability of downstream tasks highly depends on the level of cognitive abilities. We believe that our findings can provide a novel insight into the emergence of abilities in LLMs, and shed light on the future development advanced abilities of LLMs.

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

# A APPENDIX

Table 8: Calibrated accuracy (%) of all models in evaluating series. Acc and Age refer to calibrated accuracy and the age of equivalent human performance. The value of Age is calculated according to Equation 2 and rounded to the nearest integer. Bold indicates the best performance.

| Model | stage1 | | stage2 | | stage3 | | | stage4 | | | Acc | Age |
|---|---|---|---|---|---|---|---|---|---|---|---|---|
| | const | early | semio | empat | conse | induc | rever | deduc | propo | plan | | |
| OPT 125M | -16.0 | 38.6 | 26.3 | -19.0 | 16.2 | 9.3 | 4.0 | 5.6 | 4.0 | 2.2 | 7.1 | 4.8 |
| OPT 1.3B | -20.0 | 52.0 | 38.4 | -11.0 | 1.0 | 12.0 | 13.3 | -6.1 | 2.5 | -3.0 | 7.9 | 5.2 |
| OPT 2.7B | -8.0 | 53.3 | 43.7 | -9.5 | 9.4 | 12.0 | 21.1 | -1.3 | 5.5 | 3.4 | 12.95 | 6.1 |
| OPT 6.7B | -4.0 | 64.2 | 41.1 | -3.0 | 13.5 | 10.6 | 20.1 | -0.8 | -0.5 | 14.2 | 15.5 | 6.5 |
| LLaMA2-text 7B | 16.0 | 82.6 | 43.7 | -4.0 | 20.0 | 1.3 | 24.0 | -16.8 | 14.5 | 24.4 | 20.5 | 7.3 |
| LLaMA2-text 13B | 28.0 | 84.0 | 42.4 | -3.0 | 13.0 | 13.5 | 24.0 | -15.7 | 35.5 | 13.0 | 23.4 | 7.7 |
| LLaMA2-text 70B | 52.1 | 96.2 | 78.5 | 66.2 | 68.4 | 65.3 | 44.0 | 15.2 | 40.0 | 20.6 | 54.6 | 14.1 |
| LLaMA2-chat 7B | 32.0 | 78.6 | 74.5 | 54.9 | -0.2 | 41.3 | 25.3 | 6.1 | -2.0 | -0.1 | 31.04 | 10.3 |
| LLaMA2-chat 13B | 44.0 | 89.3 | 78.5 | 56.5 | 16.2 | 42.6 | 32.0 | -0.2 | 17.5 | 1.5 | 37.8 | 11.35 |
| LLaMA2-chat 70B | 52.1 | 96.2 | 78.5 | 66.2 | 68.4 | 65.3 | 44.0 | 15.2 | 40.0 | 20.6 | 54.6 | 14.1 |
| GPT-3.5-Turbo | 92.0 | 97.3 | 89.2 | 85.5 | 65.7 | 64.0 | 62.3 | 26.9 | 50.5 | 9.8 | 64.3 | 16.1 |
| GPT4 | 96.0 | 97.3 | 96.0 | **90.3** | 90.4 | 78.7 | 78.7 | 99.4 | 61.0 | 59.4 | 84.7 | 20.0 |
| Human | **100.0** | **98.0** | **96.1** | 84.2 | **98.2** | **91.6** | **92.0** | **100.0** | **82.0** | **95.6** | **93.7** | **21.5** |

Table 9: Cognitive-ability-setting-prompts of different cognitive abilities.

***Constancy***

Please imagine yourself as a child aged 0-2 years old. According to Piaget's theory of cognitive development, you are currently unable to recognize that objects exist both within and outside the field of vision and maintain a certain level of stability.

***Early Representation***

Please imagine yourself as a child aged 0-2 years old. According to Piaget's theory of cognitive development, You currently cannot give objects corresponding meanings, nor do you have a definite perception of permanent objects in the universe.

***Semiotic Function***

Please imagine yourself as a child aged 2-7 years old. According to Piaget's theory of cognitive development, You are currently unable to use symbols to represent things and concepts.

***Empathy***

Please imagine yourself as a child aged 2-7 years old. According to Piaget's theory of cognitive development, You are accustomed to thinking from your own perspective and have not yet formed a sense of empathy.

***Reversibility***

Please imagine yourself as a child aged 7-11 years old. According to Piaget's theory of cognitive development, You are currently unable to understand the reversibility of physical operations and unable to reverse thinking.

***Conservation***

Please imagine yourself as a child aged 7-11 years old. According to Piaget's theory of cognitive development, You think that external changes in form (length, shape, etc.) may affect the basic properties of an object (mass, volume, etc.).

***Induction***

Please imagine yourself as a child aged 7-11 years old. According to Piaget's theory of cognitive development, You currently cannot infer universal rules based on observed results.

***Hypothetico-Deductive***

Please imagine yourself as a teenager aged 11-18 years old. According to Piaget's theory of cognitive development, You are currently unable to deduce practical problems based on specific assumptions or rules.

***Propositional Operation***

Please imagine yourself as a teenager aged 11-18 years old. According to Piaget's theory of cognitive development, You are currently unable to understand propositions and determine the logical relationships between propositions.

***Plan***

Please imagine yourself as a teenager aged 11-18 years old. According to Piaget's theory of cognitive development, You are are currently unable to develop solutions based on specific problem.

Table 10: Examination paper to ensure the annotators are qualified.

**Question:** How many main stages are included in Jean Piaget's cognitive development theory?
**Answer:** 4

**Question:** Which stage in Piaget's theory marks the point at which children are capable of logical thinking and understanding concepts like quantity, category, space, and time?
**Answer:** Formal operational stage

**Question:** What type of operations can children perform during the concrete operational stage?
**Answer:** Addition and subtraction

**Question:** Janie knows that a bird has wings and can fly. While camping she finds a bat and thinks it's a bird, but realizes that it doesn't act the same way as a bird. She is confused. She is using what adaptation process with this new knowledge?
**Answer:** Accommodation

**Question:** What kind of activities can children engage in during the formal operational stage?
**Answer:** Abstract thinking and logical reasoning

**Question:** In Jean Piaget's cognitive development theory, which stage marks the point at which children begin to use symbols and language to represent objects?
**Answer:** Preoperational stage

**Question:** Which of the following is NOT one of Piaget's stages of cognitive development?
**Answer:** Abstract operational stage

**Question:** Children in the concrete operational stage typically understand what type of concepts?
**Answer:** Concepts of quantity and space

**Question:** What types of problems can children in the formal operational stage handle?
**Answer:** Abstract and hypothetical problems

**Question:** What are common characteristics of children in the preoperational stage?
**Answer:** Subject to egocentrism

**Question:** In the sensorimotor stage, how do children primarily explore the world?
**Answer:** Sensation and movement

**Question:** In the sensorimotor stage, how do infants primarily interact with the world?
**Answer:** Observation and sensation

**Question:** Jean Piaget's cognitive development theory primarily focuses on which age group of children?
**Answer:** Infants and children

**Question:** What types of problems can children in the concrete operational stage typically understand?
**Answer:** Logical problems

**Question:** What characteristics do children in the formal operational stage exhibit?
**Answer:** Ability to engage in abstract thinking and hypothetical reasoning

**Question:** What does Jean Piaget's cognitive development theory emphasize?
**Answer:** The active role of individuals in cognitive development

**Question:** In Jean Piaget's cognitive development theory, which stage marks the point at which children can engage in abstract thinking and hypothetical reasoning?
**Answer:** Formal operational stage

**Question:** What does Piaget's theory emphasize as influencing cognitive development?
**Answer:** A balance of social factors and genetic factors

**Question:** What can children in the formal operational stage consider when thinking?
**Answer:** Future and hypothetical situations

**Question:** What is the primary focus of the sensorimotor stage in Piaget's theory?
**Answer:** Sensory and motor exploration

