# OpenReview forum: "Tracking Cognitive Development of Large Language Models"
_ICLR.cc/2024/Conference — ICLR 2024 Conference Withdrawn Submission_

### Official Review · Reviewer_Fc8y · 2023-10-29

**Soundness:** 2 fair
**Presentation:** 3 good
**Contribution:** 1 poor
**Rating:** 3
**Confidence:** 5

**Summary:**

To reveal and evaluate the cognitive abilities of LLMs, this paper introduces Piaget’s Theory of Cognitive Development (PTC) in psychology and construct a corresponding benchmark (CogLM). Through extensive experiments on various LLMs, it shows some interesting findings and provides guidance for the future development advanced abilities of LLMs.

**Strengths:**

The paper is well organized and the ideas are skillfully presented. However, this paper is not innovative enough and the key content is not clear enough. As far as I am concerned, the main advantages are as follows.

1. The paper is easy to follow and understand. No matter the motivations, methods, experiments, and analysis in this paper, they are all flowing and graceful.

2. Some findings and conclusions in this paper are interesting and meaningful. For example, “cognitive abilities are inherent to the LLMs and cannot be enhanced through multi-step reasoning.”, “the parameter size of LLMs can be analogous to a child’s brain capacity, and cognitive abilities continuously improve with the development of the brain”.

3. Although it is not the first time that human evaluation methods are applied to LLMs and some details are not clearly explained, the solution seems to be effective and reasonable. This paper is able to evaluate the model from the perspective of psychology and provide some future development suggestions for LLMs.

**Weaknesses:**

I still have some concerns which, in my opinion, are the weaknesses:

1. I am skeptical about using PTC to evaluate LLMs. The paper needs to further emphasize whether it is reasonable to use PTC to evaluate the model, or give some basis or experimental results. More about the deep introduction of PTC in psychology and its correlation with model evaluation need further elaboration, otherwise it will obviously not be able to convince readers by directly applying human evaluation datasets to the LLMs’ evaluations.

2. The expressions such as “tracking”, “different stages”, “evolution process” in the paper are confusing. What exactly are the different stages of LLM? The paper mentions that "PTC suggests that intelligence grows and develops through a series of stages." However, the LLM in the paper is fixed and does not “grow” continuously. I think it would be best to truly evaluate the LLM at different stages. For example, can different amounts of data using in the pre-training and fine-tuning processes be regarded as different stages? It is more appropriate to apply PTC to provide evaluation services along the LLM growth cycle.


3. The benchmark CogLM constructed based on PTC in this paper only uses multiple-choice questions as the assessment format. Is the generative ability also a manifestation of the LLMs’ "cognitive ability"? I believe that most human cognitive testing programs are more about answering each test question fluently than multiple-choice questions. The model's complete response is clearly more informative than the selected options.

**Questions:**

See above.

---

### Official Review · Reviewer_BFZ5 · 2023-10-29

**Soundness:** 1 poor
**Presentation:** 2 fair
**Contribution:** 1 poor
**Rating:** 1
**Confidence:** 5

**Summary:**

This paper focuses on an assessment of LLMs’ cognitive levels. The authors use Piaget’s Theory of Cognitive Development (PTC) as a tool to evaluate the "cognitive abilities" of various generative LLMs, and construct a benchmark test suite (CogLM) for this purpose. CogLM demonstrates the stages of PTC, except the first stage of PTC has not been considered (CogLM excludes reflexes and sensorimotor aspects of multimodal interaction). Multiple LLMs such as OPT, LLaMA2-chat-base, GPT-3.5-Turbo, and GPT-4 have been evaluated on CogLM. CogLM consists of questions in a multiple-choice format. Part of paper attempts to assess the consistency of CogLM with PTC and ensure the alignment between CogLM and PTC. In order to do so, humans between the age of 6 and 20 have been recruited to respond to CogLM questions. Spearman and Pearson correlation coefficients between the age of participants and the questionnaire scores have been computed to validate the effectiveness of the standard annotation guidelines.  Furthermore, the mapping function between the accuracy of response and age of participants have been computed through regression. The authors claim to show that GPT-4 exhibits human like cognitive abilities commensurate with a 20-year-old human.

**Strengths:**

The CogLM benchmark is novel and would likely to be valuable to the community for different evaluations and assessments.  I appreciate the level of work that must have gone into creating it.  There is a wide variety of experiments performed and results presented in a fairly clear and concise fashion.

**Weaknesses:**

Unfortunately there are a number of serious, I would say fatal, flaws to the paper, in that the entire endeavor rests on faulty assumptions and a non-trivial number of false statements.  The authors seem to set out with the goal to demonstrate that LLMs have human-level "cognitive" abilities and then construct an experimental protocol in order to arrive at that conclusion, which is begging the question.

1) There is a fundamental category error being made in the statement that the abilities demonstrated by LLMs are fundamentally cognitive.  LLMs are sophisticated word-prediction machine, and this capacity happens to provide ability in tasks like summarization and question answering.  The statement "However, there is little theoretical evidence regarding why and how this performance has been achieved" is frankly not true.  There is plenty of theorizing that the ability to learn statistical patterns in language is precisely why LLMs are able to reproduce human-like outputs. Due to the large amount of data and large model size, they are good predictors of what a human would say, but doing so does not mean the same cognitive processes are at play.  A fairly simple neural net can accurately predict how a human would move in a given task.  It doesn't mean that network has the same sensorimotor processes as a human.  Prediction is not experience.  In general, the underlying definition of “cognitive abilities” being used here is faulty because LLMs are not cognitive systems, but rather statistical models based on sophisticated word occurrence correlation, to say nothing of studies of GPT’s varying ability over time: e.g.,
Chen, L., Zaharia, M., & Zou, J. (2023). *How is ChatGPT's behavior changing over time?*. arXiv preprint arXiv:2307.09009.

2) Speaking of sensorimotor, by excluding the reflex and sensorimotor aspects of multimodal interaction with the world destroys the scaffold upon which successive Piagetian stages are built.  Piaget’s theory is that the development of the different stages are supervenient upon passing through the previous stages.  How can an LLM have achieved a certain level of cognitive functioning according to Piagetian theory if it was trained to that level without passing through the previous stages first?  Without sensorimotor capability, and LLM lacks the mechanism to experience the sensorimotor stage. This invalidates the entire construct of using PTC as a measure of cognitive ability, as the foundation has been pulled out.  This renders the entire comparison invalid, and makes PTC an inapprorpriate evaluation tool, since the entire first stage of PTC is explicitly the sensorimotor phase and everything else is scaffolded on top of that.

3) Piaget is well-known but also quite contentious within dev-psych circles. Piagetian theory was based primarily on case studies and only some (not all) of his ideas have been supported through experimental methodologies.  This makes PTC in general a dubious metric by which to assess cognitive ability.

4) GPT-2 is not similar to a human infant.  GPT-2's vocabulary is far larger than an infant but its ability to produce coherent, grammatical, on-topic, contextually-correct text is inferior to more recent models, as well as to toddlers given a toddler's vocabulary. GPT-2 is still an autoregressive next word predictor, explicitly trained for generation over text mostly written by adults.  This is not how human children learn language.  The fact that the data and compute power available at the time rendered GPT-2 inferior to GPT-4 does not make it at all like a human infant.  Did the authors forget how state of the art GPT-2 was considered at the time of its release?

5) The benchmark is all multiple choice questions.  This would be fine if humans only thought in multiple choice questions but we know that is obviously not the case.

6) In Section 3.4, the definition of accuracy is changed between construction of the benchmark and using it for LLM evaluation.  "After confirming the positive correlation between answer accuracy and cognitive age, we aim to further construct the mapping function between them. We first make adjustments to the method of calculating accuracy."  After demonstrating a correlation between accuracy and cognitive age, the authors then change how they calculate accuracy?  The authors seem to be in search of a result that fits their narrative that LLMs are somehow following a human-like evolutionary path.

7) An assertion in Section 4.2: "Overall, the cognitive abilities of the OPT, LLaMA2-chat 70B, GPT-3.5-Turbo, and GPT4 models successively increase, and the performance of each model gradually declines with the increase of stage, consistent with humans." is not self-consistent, and the assertion that human cognitive performance declines as they progress through PTC stages, is also not true.

8) The authors claim that GPT-3.5 and GPT-4 surpass humans at empathy.  Empathy is the ability to understand feelings and emotions.  Even if we allow that LLMs are accurate reproductions of the language faculty in the brain, emotions are governed by different regions and mechanisms.  LLMs can no more understand emotions than ELIZA could.

9) Table 5 claims that the results are demonstrations of human-like cognitive abilities but these number measure the ability to correctly answer multiple choice questions.  The ability to answer multiple choice questions is not in anyway a global indicator of cognitive ability at the “ages” being tested.

**Questions:**

1) "As the most authoritative theory in the development of psychology" (should "the development of" be "development*al*."  Piaget is not as influential in other branches of psychology outside dev-psych.

2) "Out of the 207 completed questionnaires, 141 are deemed valid (based on the reasonableness of test duration)".  What is "reasonableness of test duration"?  This is a qualitative and unjustified assertion.

3) How could a human in PTC's first stage (0-2 years old) answer questions in language like those presented in Table 1?

4) In general, I wonder what the point is of doing a simple performance-based evaluation of evaluating anything on GPT-4, given that we know nothing about its data or training.

5) "For text-completion models, as they lack the ability to follow instructions and their output format is difficult to control, we concatenate each option with the corresponding question as input, and take the option with the highest generation probability as the model’s prediction." So the input is different for the different models, therefore the input has to be tuned (how so? manually?) to extract the “best” output format from the different models. The input format is not controlled.

6) Why would accuracy be the important metric for most of the factors evaluated (as in Table 4)?

7) "We explore this question from two perspectives: the parameter size and the optimization objective of LLMs, as they are the most natural assumptions."  Why are they the most natural assumptions?

8) Table 5: 1) No particular approach comes out on top. Are these results supposed to show anything about a general purpose capacity of LLMs?

---

### Official Review · Reviewer_sSK2 · 2023-11-01

**Soundness:** 1 poor
**Presentation:** 2 fair
**Contribution:** 2 fair
**Rating:** 3
**Confidence:** 4

**Summary:**

The paper proposes a new dataset for evaluating LLM cognitive capacity. The dataset is inspired by Piaget’s Theory
of Cognitive Development (PTC) which is highly influential in developmental psychology. The authors design language-based tasks that target skills developed at different stages of human life, validate their data on human participants, and create a conversion procedure allowing to estimate one's cognitive development age based on their task performance. They evaluate a number of LLMs on the newly proposed tasks, showing that generally, more parameters and RLHF-fine-tuning corresponds to higher cognitive age.

**Strengths:**

I highly resonate with the goals of the paper. Given extremely fast advances in LLM performance, it is crucial to devise thoughtfully designed and thoroughly validated datasets to better understand the capacity and limitations of LLM cognitive abilities.

I believe that this work, therefore, tackles an extremely important topic that is of high interest to a large portion of AI, ML, and Cognitive Science communities.

What further strengthens the paper is that they based their assessment tool on a highly established theory of cognitive development, which adds additional credibility to the research.

**Weaknesses:**

## Quality

The paper creates a very uneven impression when it comes to the quality of the experimental support of its claims. On the one hand, I find the experiments interesting, and I deeply appreciate that the authors include human evaluation of the generated dataset.

On the other hand, there is a number of substantial limitations:

### Weakly justified design choices

Many of the design choices seem to be poorly justified or not justified at all. For example, the choice of the calibration function is not explained nor is it compared to alternatives (e.g. F-measure). In this context, it's also not clear why the number of answer options was not standardized in the first place. For example, some questions have very restrictive answer options "Yes" and "No" while others are more open-ended and have more options. This situation could have been easily avoided by changing question-writer instructions.

### Statistical reporting

I also believe that statistical result reporting could be improved. For example, on page five, in the first paragraph, the authors say "We observe that w1 : w2 : w3 : w4 = 1 : 2.6 : 1.4 : 2.5, indicating that cognitive abilities in the second and fourth stages are better at reflecting cognitive age under the evaluation of CogLM". Such hypotheses need to be directly tested on the available data. Without uncertainty estimates on these parameters, it's impossible to gauge the reliability of this conclusion. Since this is a result of secondary importance, I would suggest removing this phrase entirely or adding appropriate statistical tests (e.g. method of contrasts, and, before that, comparing the restricted model (with all coefficients equal) and unrestricted models to show that these differences between w_is are significant).

### Speculative claims

I find some claims to be highly speculative and not supported by the data. For example,

"Specifically, the cognitive abilities of LLMs continuously improve as the size of model parameters increases, which follows the same pattern as human cognitive abilities continuously improving with age. We hypothesize that the reason for this phenomenon is that the parameter size of LLMs can be analogous to a child’s brain capacity, and cognitive abilities continuously improve with the development of the brain, as stated in Piaget et al. (1952)."

The paper does not go into sufficient depth to rigorously define terms like "child's brain capacity". As a result, the claims like above seem highly speculative and/or vacuous. If we condense the claim above, removing terms that are hard to define, it simply says that "higher parameter count is associated with better cognitive task performance", which we already know since all standard model evaluation tasks are cognitive tasks as well.

I believe that it's better to restrict the amount of such claims as much as possible, as it hurts the credibility of the paper.

In general, the pattern observed in the paper can be formulated as "better models tend to perform better on the proposed task". The paper does relatively little to establish the correspondence between model development and human cognitive development.

To clarify what I mean - in order to show that LLMs actually follow a path similar to human development, one would need to demonstrate something that goes beyond the general pattern of better models performing better on a new task. The paper has some results of that kind - specifically, tasks that correspond to later stages in cognitive development seem to be generally harder for LLMs.

It's crucial, however, to look into alternative explanations, one of which is that these tasks might also be harder to generate in unambiguous ways. In other words, we need to validate the data and see (adult) human accuracies on tasks corresponding to different stages of development. This is especially important given that the authors found out that a large proportion of initially generated questionnaires was not valid (i.e. the data generation process is quite noisy).

In a similar vein, I do not fully agree with the statement "3" in the abstract:

"(3) The ability of downstream tasks highly depends on the level of cognitive abilities." - I don't believe that the authors showed the causal direction. I.e. performance on downstream tasks correlates with their cognitive abilities, as measured by the authors, but the claim that there is a causal relationship is much stronger, and would require a lot of additional experimentation and analyses.



### "Ability erasure" experiments

While I find the idea interesting, I find the experiments deeply flawed and presentation slightly misleading. While in the main paper, the authors said that the model was prompted with phrases like "you have not yet developed a sense of empathy", in fact the prompts were very different (as given in the appendix):

For example:
"Please imagine yourself as a child aged 0-2 years old. According to Piaget’s theory of cognitive development, you are currently unable to recognize that objects exist both within and outside the field of vision and maintain a certain level of stability."

Given this prompt, the apparent results that we see (later abilities "depending" on earlier ones) are explained by the model being able to imitate the behavior of a human of the requested age. I.e. the authors don't say "do everything as usual, but pretend that you don't have empathy" to surgically remove empathy, but rather ask the model to imitate a person at a given age, specifically requesting it to conform to the Piaget's theory.

In other words, the authors (inadvertently) directly asked the model to generate the results that they observed.

### Insufficient data validation

In general, given how much the paper's claims depend on the quality of the generated data, I find the data validation experiments insufficient. We do see that age correlates with performance on the proposed dataset, but I believe that a much more detailed look into human performance is warranted (in the very least, splitting humans by age and showing how they perform on different subtasks).

## Clarity

Unfortunately, the paper is not very clearly written, and in general, the presentation can be substantially improved.

For example, consider this sentence in the section 3.3
"Out of the 207 completed questionnaires, 141 are deemed valid (based on the reasonableness of test duration)."
"Duration" was never mentioned before that snippet, and is never mentioned after. It is also not clear what criterion was used to see whether a questionnaire was "valid".

Unfortunately, this is not the only case when the paper does not fully define its terms or procedures.

## Typos & Phrasing suggestions

"recently challenging tasks have been proposed" -> "recently, challenging tasks have been proposed"

"Theory of Cognitive Development (PTC) is the most authoritative theory in the development of psychology, developed by Jean Piaget" -> perhaps should be rephrased to avoid three-fold repetition of the word "development". More importantly, it should, perhaps, be "developmental psychology", not "development of psychology".

"cognitive abilities of PTC" -> "cognitive abilities proposed by PTC"

"the ability of downstream tasks" -> "the ability of [performing?] downstream tasks"


# Conclusion

Overall, I do highly resonate with the theme of the paper and I find many of the ideas promising. At the same time, unfortunately, I believe that there are serious flaws with the experimental design, substantial inaccuracies when it comes to result interpretation, and (less importantly), presentation clarity concerns. At present, I can not recommend its acceptance.

At the same time, I believe that the paper has most of the elements it needs to be published in a good venue. I deeply hope that the authors will rework the framing of the paper, scaling down some of the claims, and/or adding new experiments to get a more thorough comparison between human and LLM development.

**Questions:**

### Lack of actual LLM developmental data

The authors make an analogy between human development and fully-trained LLMs of different capacity. At the same time, for humans, the process is fundamentally different - as the brain develops, it realizes its potential rather than simply increases in capacity. A more natural analogy, therefore, seems to be a single LLM evaluated on different training stages.

It would highly increase the impact of the paper if the authors evaluated LLM cognitive task performance at different stages of training. I understand that for some models, such an experiment would not be feasible. But perhaps it might be possible to find open-source model training checkpoints to evaluate the model on these tasks in different stages of training.

---

### Official Review · Reviewer_LvrM · 2023-11-02

**Soundness:** 2 fair
**Presentation:** 3 good
**Contribution:** 2 fair
**Rating:** 3
**Confidence:** 4

**Summary:**

This paper adopted a cognitive framework from psychology (PTC framework) to evaluate LLMs. The authors created a human-annotated benchmark dataset based that contains questions from 10 cognitive levels and 4 cognitive stages. The number of questions increases with cognitive difficulty levels. The questions are formatted as multiple choice questions and the main evaluation metric for this dataset is calibrated accuracy. The authors benchmarked various popular language models on this dataset, showing that the current best model GPT-4 has a similar cognitive ability of a 16-year-old human. This paper provided some additional analysis of LLMs' cognitive abilities. However, the analysis was not convincing to me personally. I will elaborate on the details in the following sections.

**Strengths:**

Benchmarking LLMs (or general AI models) on human-level tasks is an important topic. Besides, this paper constructed an expert-annotated benchmark dataset, which could be useful to the research community.

**Weaknesses:**

Even though the resources of this paper could be useful, the analysis of the CogLM benchmark could be improved. From Section 4.3.1,  the two findings are LLMs with larger parameter size have better cognitive abilities and optimization objective significantly improves cognitive abilities. However, the authors reached the latter conclusion by comparing the cognitive abilities of llama-2-text and llama-2-chat. However, the differences between these two models are not just optimization objectives (word prediction vs. RLHF). The llama-2-chat was finetuned on a large amount of labeled instruction data. Whereas the main motivation for RLHF of LLMs is to align the values and preferences of humans. I believe is inconclusive to say the differences in cognitive abilities are mainly from the optimization objective, and the improvements could be resulted from the instruction tuning with the same language modeling objective.


In 4.3.2, the authors mentioned that CoT does not bring significant performance improvement in terms of average calibrated accuracy, and hypothesized that cognitive skills are inherent and cannot be improved by CoT. However, from Table 5, we can see CoT’s effects vary according to different cognitive levels, and CoT clearly showed a performance improvement in planning and propositional operations. This is expected as CoT itself was proposed to solve problems of a more complex nature. Hence, using the average result of all cognitive levels is questionable in analyzing CoT.
Besides, from Table 4 and Table 5, we can see the results of GPT3.5-turbo in Table 4 correspond to the row ‘With CoT’ in Table 5 instead of ‘Base’. This raised a serious question, so are all results in Table 4 are prompted with CoT? If so, why there is no description of this at all? CoT only showed up in section 4.3.2 of this paper. The authors need to double-check whether Table 4’s results are “with CoT” or GPT3.5-turbo’s result in Table 4 should be ‘Base’.


On the clarity side, the notation of ‘chain-of-cognition’ is highly confusing. This term refers to ‘chain-of-thought’ in Section 4.3.2 and somehow became “CoC” in Section 4.3.4 and Table 7. Simply coming up with some terms does not add novelty to this work but only adds confusion to readers.

**Questions:**

From Table 4 and Table 5, we can see the results of GPT3.5-turbo in Table 4 are corresponding to the row ‘With CoT’ in Table 5 instead of ‘Base’. This raised a serious question, so are all results in Table 4 are prompted with CoT?

Why do you need to introduce ‘chain-of-cognition’ when it essentially refers to CoT?